# Artificial Intelligence and Advanced Melanoma: Treatment Management Implications

**DOI:** 10.3390/cells11243965

**Published:** 2022-12-08

**Authors:** Antonino Guerrisi, Italia Falcone, Fabio Valenti, Marco Rao, Enzo Gallo, Sara Ungania, Maria Teresa Maccallini, Maurizio Fanciulli, Pasquale Frascione, Aldo Morrone, Mauro Caterino

**Affiliations:** 1Radiology and Diagnostic Imaging Unit, Department of Clinical and Dermatological Research, San Gallicano Dermatological Institute IRCCS, 00144 Rome, Italy; 2SAFU, Department of Research, Advanced Diagnostics, and Technological Innovation, IRCCS-Regina Elena National Cancer Institute, 00144 Rome, Italy; 3UOC Oncological Translational Research, IRCCS-Regina Elena National Cancer Institute, 00144 Rome, Italy; 4Enea-FSN-TECFIS-APAM, C.R. Frascati, via Enrico Fermi, 45, 00146 Rome, Italy; 5Pathology Unit, IRCCS-Regina Elena National Cancer Institute, 00144 Rome, Italy; 6Medical Physics and Expert Systems Laboratory, Department of Research and Advanced Technologies, IRCCS-Regina Elena Institute, 00144 Rome, Italy; 7Departement of Clinical and Molecular Medicine, Università La Sapienza di Roma, 00185 Rome, Italy; 8Oncologic and Preventative Dermatology, IFO-San Gallicano Dermatological Institute-IRCCS, 00144 Rome, Italy; 9Scientific Direction, San Gallicano Dermatological Institute IRCCS, 00144 Rome, Italy

**Keywords:** metastatic melanoma, targeted therapy, immunotherapy, artificial intelligence, precision medicine

## Abstract

Artificial intelligence (AI), a field of research in which computers are applied to mimic humans, is continuously expanding and influencing many aspects of our lives. From electric cars to search motors, AI helps us manage our daily lives by simplifying functions and activities that would be more complex otherwise. Even in the medical field, and specifically in oncology, many studies in recent years have highlighted the possible helping role that AI could play in clinical and therapeutic patient management. In specific contexts, clinical decisions are supported by “intelligent” machines and the development of specific softwares that assist the specialist in the management of the oncology patient. Melanoma, a highly heterogeneous disease influenced by several genetic and environmental factors, to date is still difficult to manage clinically in its advanced stages. Therapies often fail, due to the establishment of intrinsic or secondary resistance, making clinical decisions complex. In this sense, although much work still needs to be conducted, numerous evidence shows that AI (through the processing of large available data) could positively influence the management of the patient with advanced melanoma, helping the clinician in the most favorable therapeutic choice and avoiding unnecessary treatments that are sure to fail. In this review, the most recent applications of AI in melanoma will be described, focusing especially on the possible finding of this field in the management of drug treatments.

## 1. Introduction

In recent decades, new therapeutic perspectives for cancer treatment have shifted toward precision medicine, personalized to patient characteristics. The continuous discovery of new molecular markers and the use of innovative techniques make possible a more delineated view of the tumor and less harmful treatments [1]. The molecular and biochemical analyses and high-resolution medical images are a precious source of information that are not always easy to decode and use. Artificial intelligence (AI) allows one to amalgamate all these data to obtain new information about the possible prognosis and patients treatment [2]. Systems based on AI are entering the medical field at an impressive speed, assisting the clinician in choosing the most appropriate therapies with the highest success rate [3]. Clearly, this leads to better management of the oncology patient in terms of quality of life and cost reduction. Metastatic melanoma is, certainly, one of the pathologies that seems to benefit from modern mathematical and computational data processing approaches. Some interesting studies, indeed, have shown that the use of appropriate algorithms has allowed a greater and significant accuracy in the diagnosis of melanoma than the clinician’s experience alone, contributing to earlier diagnoses [4,5]. These results are particularly encouraging because the metastatic melanoma still represents, today, a particularly aggressive form of cancer characterized by unfavorable prognosis. Melanoma is characterized by highly heterogeneous properties and is supported by many genetic alterations and by a microenvironment highly favorable to the development of metastases, even in distal sites [6]. If not recognized at an early stage, melanoma is the skin cancer with the highest mortality rate [7]. In addition, melanoma is characterized by high intertumoral and intratumoral heterogeneity. This implies that clones of the same tumor may harbor different mutations and originate different metastases from the primary lesion. All this, of course, results in complex diagnosis and less accurate treatment. The need for a personalized approach in treating the melanoma patient requires a deep understanding of intratumoral and intertumoral heterogeneity at the genomic, transcriptomic, and proteomic levels [8]. In this context, the application of AI technologies can improve and standardize the management of the melanoma patient by the clinician. Indeed, AI (encompassing computer science and technology) is a rapidly evolving field that is revolutionizing many aspects of our lives, including the medical area. Through the ability to interpret and relate a wide range of information, AI allows a more complete view of the disease and all possible therapeutic scenarios. Obviously, this also applies to complex and multifactorial pathologies such as melanoma.

In this review, we will focus particular attention on the new AI applications for the melanoma patient management. Indeed, we will describe, specifically, how new technologies can help clinicians in therapeutic choices (targeted therapy and immunotherapy) and in the randomization of patients.

## 2. New Frontiers in Melanoma Treatments: An Overview

Metastatic melanoma treatment has undergone profound changes in recent decades, thanks mainly to the introduction of molecular therapies and immunotherapy into the clinical setting [9]. These treatments, which are specific against only tumor cells, have occupied more and more space in the therapies of patients with melanoma, supplanting, in many cases, chemotherapy that is not very selective and has numerous side effects. Progress in survival and quality of life has been very satisfactory, but the phenomenon of treatment resistance has not yet been resolved [6]. A large percentage of patients, indeed, develop resistance to therapies and undergo disease progression. Thus, the identification of new biomarkers (molecular or imaging) that are potential targets for new therapies or prognostic factors of response to treatments is to be hoped for.

### 2.1. Targeted Therapy

Mitogen-activated protein kinase (MAPK) is the most dysregulated pathway in melanoma, and about 50–60% of all melanoma’s somatic mutations are located at the v-Raf murine sarcoma viral oncogene homolog B (BRAF), where valine replaced by a residue of glutamic acid (V600E) is the most representative [10]. In this context, BRAF represents the standard therapeutic targets, and several preclinical studies have shown that its blockade allows a reduced cell growth and induction of apoptotic process [10,11,12]. Vemurafenib (or PLX4032), dabrafenib (or GSK2118436), and encorafenib (or LGX818), approved by the Food and Drug Administration (FDA) in 2011, 2013, and 2018, respectively, have significantly improved patients’ life expectancy, both in terms of overall survival (OS) and quality of life, compared to the chemotherapy [13,14,15]. Moreover, the therapeutic combination between BRAF and mitogen-activated protein kinase (MEK) inhibitors (cobimetinib/trametinib/binimetinib) promotes, for metastatic melanoma, a more durable response to treatments over time, preventing the paradoxical MAPK pathway re-activation [16,17,18,19].

### 2.2. Immunotherapy

Melanoma, in its primary form, is characterized by the presence of a relevant lymphocytic component and, for this reason, the immunotherapy has produced comforting results in terms of survival. Although several drugs are in the trials, only the immune checkpoint inhibitors (ICIs) have been approved and used in the clinical setting for the melanoma treatment (Table 1).

ICIs are monoclonal antibodies developed against specific targets, such as T-lymphocyte antigen 4 (CTLA-4), programmed cell death protein 1 (PD-1), and PD-1 ligand (PDL-1) and are designed to eliminate the blockage of T-cell activity against tumors [28,29]. Ipilimumab (MDX-010) is the most important specific monoclonal antibody against CTLA-4 approved by FDA and the European Medicines Agency (EMA) in 2011, alone or in combination with PD-1 inhibitors. Several melanoma clinical trials have shown that, compared with chemotherapy, ipililumab results in better OS outcomes [30,31,32]. PD-1/PDL-1 axis represents a key therapeutic target on which several drugs have been developed [33,34,35]. In the metastatic melanoma treatment, two anti-PD-1 drugs have been approved in 2014 by the FDA: nivolumab (BMS-936558, MDX-1106) and pembrolizumab (MK-3475). Both are involved in T-cell function reactivation by blocking the PD-1/PDL-1 interaction, and several studies have shown that, alone or in combination with ipilimumab, they promote better results in terms of OS in metastatic melanoma patients [36,37,38,39]. Several molecules relating to anti PDL-1 have also been developed: BMS-936559 (MDX-1105) is involved in inhibition of the binding between PDL-1 and its receptor; atezolizumab (MPDL3280A), durvalumab (MEDI4736) and avelumab (MSB0010718C), instead, are related to three antibodies with high affinity and specificity to PDL-1, which are involved in several clinical trials for melanoma metastatic treatment [40].

## 3. Artificial Intelligence in Oncology

Artificial intelligence (AI), developed since 1956, is a field of informatics that through complex systems, allows machines to learn from experience, mimicking the capabilities of the human mind [41]. In particular, AI is able to process and manipulate a large amount of data and extrapolate correlations between them in a way that the human brain would not recognize (Figure 1). AI’s application fields are innumerable and have revolutionized and improved the management of many activities [42]. Medicine, too, in its many aspects, is now exploiting the great potential of AI to improve health care delivery, clinical decision-making, and delineate patient treatment [43,44,45].

Machine learning (ML) is focused on the creation of AI programs that learn or improve performance based on the data they use. It enables algorithm creation that can learn and make predictions. In the medical field, especially for oncological pathologies, ML has allowed experts to produce personalized treatment predictions by using and schematizing, in a rigorous way, databases obtained from many patients. All this has resulted, of course, in a better approach to decision-making for the clinicians [46]. The two main machine learning algorithms used are supervised and unsupervised, and their difference is essentially determined by how each algorithm learns the data to make predictions [46]. Supervised ML is certainly the most frequently used algorithm and is characterized by the intervention of an operator who “teaches” the machine using data sets already available. Unsupervised ML, instead, provides the ability of the machine to learn complex models without necessary constant intervention on the part of an operator. A third category of algorithms is characterized by reinforcement learning, when the machine independently apprehends from the environment without being taught explicitly [47]. Deep learning (DL) is an algorithm type that uses a multi-layer neural network architecture to learn representations of data automatically [48]. It simulates the human brain in processing and the sub-division of complex data into interconnected steps. In recent years, DL has also been under constant development in the medical field and will become essential for predicting drug response and optimizing drug dosages. All this will be possible through the analysis and interpretation of complex molecular, biochemical, and clinical datasets [49]. As mentioned above, the AI application areas are many, and all are directed to simplify our lives and decisions. In recent decades, the medical world has also increasingly relied on AI to assist in the interpretation of data and to improve performance and clinical decisions [50]. AI has certainly found fertile ground in oncology, where it has enabled improved interpretation of clinical images and subsequent diagnoses. A striking example was the approval by the FDA of two softwares able to quickly process mammogram images for the early detection of breast cancer and to identify colon cancer [51,52,53]. Diagnosis, prognosis, and treatment of the oncological patient, specifically the melanoma patient, involves the interplay of information derived from different omics sciences (genomics, transcriptomics, radiomics, etc.) [1]. In this intricate background, AI plays a key role in the interpretation and management of these multiple data and contributes to precision medicine as new frontier in the treatment of the cancer patient. In recent years, the discovery that the tumor is not an “isolated” mass, but an elaborate system that interacts with the surrounding microenvironment, has established the basis for an important concept: each patient has different characteristics that drive different responses to therapies [1]. Melanoma is certainly one of the cancer forms in which precision medicine is most relevant. Indeed, it is an extremely complex disease characterized by many genetic mutations and by an immune microenvironment that favors drug resistance and disease progression [6]. For melanoma, but not only melanoma, AI is greatly assisting the clinicians in their therapeutic choices, as it has made it possible to rapidly study, process, and analyze personal variability in response to treatments [54,55,56]. A tumor’s qualitative and quantitative analysis goes through the study of multiple genetic, molecular, and biochemical features. All of these allow adaptation and/or improvement of treatments for the individual patient while also predicting, where possible, the efficacy of therapies. In melanoma, an example is the preliminary assessment of BRAF status, an important indicator of targeted therapy response [57].

### AI and Radiomics

The current development of omics sciences (genomics, transcriptomics, proteomics, metabolomics, and radiomics) has resulted in detailed knowledge of pathways, microenvironment-tumor interactions, genetic alterations, etc. that contribute to failure and/or response to cancer therapies [1]. Amongst all omics sciences, radiomics represents a field that is rapidly expanding and promises to be a valuable tool in the identification of imaging biomarkers contributing to precision medicine (both in terms of prevention and treatment) [58]. This new field of medicine is, therefore, responsible for translating radiological images into quantitative data on which to obtain biological information. This is possible thanks to AI, which, by exploiting the massive imaging datasets, can train diagnostic and prognostic models for a variety of tumor types [55,59,60]. In recent years, many radiomics studies have been developed with the primary purpose of improving the management of the cancer patient. In the melanoma context, a large part of the studies involved therapy management through radiological identification of predictive biomarkers in treatment response. A recent study analyzed the predictive role of radiomic analysis of magnetic resonance images (MRI) in response to immunotherapy in patients with melanoma brain metastases. The authors, after drawing the volumes of interest, performed univariate Cox regression for each radiomic feature analyzed followed by Lasso regression and multivariate analysis. They retrospectively analyzed MRI images of 196 patients with melanoma brain metastases and observed that higher-order MRI radiomic features were associated with a better response to treatments [61]. Basler and collaborators [62] have, on the other hand, highlighted the potential predictive role of radiomics, in association with biological markers, in pseudoprogression of disease in patients with melanoma and treated with immunotherapy. They, indeed, observed in 112 patients with metastatic melanoma, that noninvasive PET/CT-based radiomics and LDH/S100 are, in combination, good markers for detecting disease progression within a three-month time frame. This information obtained can be exploited to delay or anticipate immunotherapy treatments. However, several other studies have combined radiomic analysis with biological markers to potentiate the tumor information. Recently, for example, radiomic imaging of 15 patients with metastatic melanoma was combined with circulating tumor DNA (ctDNA) to assess disease progression. This study suggests the possibility of combining radiomics and liquid biomarkers to analyze changes in tumor characteristics [63]. An interesting study investigated the possible correlation between texture analysis on pre-treatment contrast-enhanced computed tomography (CT) images and OS and progression-free survival (PFS) in patients with metastatic melanoma treated with pembrolizumab (monoclonal antibody anti-PD-1). The retrospective study was conducted on 31 patients undergoing treatment with pembrolizumab, and a total of 74 metastatic lesions were analyzed. AI analyses conducted showed that tumor skewness derived from pre-treatment CT texture analysis can be considered a good predictive biomarker of OS and PFS for metastatic melanoma treated with pembrolizumab. Indeed, skewness values greater than -0.55 were significantly associated with both lower OS and lower PFS after pembrolizumab administration [60]. In general, AI is responsible for analyzing and cataloging the multitude of data obtained in a manner that human intervention alone cannot do. The use of specific algorithms has resulted in the identification of new biomarkers, as well as has driven gene sets in cancer and the subsequent design of new drugs [64] (Figure 2).

In oncology, disease progression is a highly complex process because many factors, which can affect or not affect the cancer cells’ transformation, are involved. It is, therefore, difficult for the clinician to specifically delineate a disease course. One AI application example finds fertile ground in this particular field, where the application, analysis, and study of complex mathematical system can predict whether or not the disease will evolve. A very interesting work that extensively investigated this aspect of AI was performed by Jeffrey West and collaborators in 2016 [65]. The authors of the study showed that early tumor analysis (subclinical phase), when the tumor mass is not yet visible, could help in understanding the metastatic process, which is still obscure in many ways today. Studying the early interactions of heterogeneous cell populations and how they compete for their surroundings could give a picture of how tumor progression develops before the tumor is clinically detectable. To understand how a tumor acts before it is detectable, a mathematical model was used to analyze the relationships between tumor and healthy cells competing for dominance. The authors, following an evolutionary game from the 1950s called the “Prisoner’s Dilemma”, assigned payoffs to each cell population, considering healthy cells cooperators while cancer cell defect. From the analysis of various aspects, such as death rate, point mutations, heredity, etc., the study showed, through mathematical analysis, that natural selection can push toward contexts in which the cancerous aspect prevails. The analysis of how malignant cells prevail and the trend of tumor progression, of course, could bring crucial information for the correct drug treatments. Therapies started in the early stages of tumor mass development certainly show better responses than those undertaken in the later stages, as described by simulated drug strategies and therapeutic response. A very recent study, on the other hand, used coordination games to assess and understand cellular interactions. The authors proposed some aspects of cancer as results of coordination games and exploited the results for better coordinate therapies. According to this strategy, cell populations playing a coordination game converge on a predominant phenotype, eliminating all competing ones. According to the authors, this can be exploited for the setting of proper therapy, as clinicians could design treatment plans aimed at keeping tumor size and composition under control. Obviously, when two cellular phenotypes compete, new therapeutic regimens can be designed to maintain heterogeneous tumor composition [66].

## 4. Metastatic Melanoma Management: Artificial Intelligence Implications

The clinical choice to perform a specific treatment involves the oncologist’s analysis and study of many factors, such as disease grade, mutational status and, of course, the patient’s condition. Such choices, although linked to standard treatment regimens, are often difficult to interpret in many cancer contexts, such as metastatic melanoma. Therefore, the constant development of AI methods, applicable to the medical area, capable of interpreting and analyzing a massive amount of data, could help the clinician in the management of the melanoma patient and the most appropriate therapeutic choice. In addition, such methods could help in the prediction of possible disease recurrence and response to standard treatments, thus hypothesizing different treatment scenarios [67]. Our focus is to represent the most interesting and recent studies in which AI has been used to improve the management of the patient with advanced melanoma.

Goussalt et al. [68] have developed and validated four machine learning models to predict the response to immunotherapy and targeted therapy in stage IIIc or IV melanoma patients. The work was conducted on data from 10 centers participating in the French network for Research and Clinical Investigation on Melanoma (RIC-Mel), launched in 2012, with about 935 patients, corresponding to 1978 systemic treatments have been extracted from RIC-Mel database. Several data were considered in this study, including age, sex, melanoma type, spontaneous regression, number of invaded lymph nodes, extracapsular extension, mutational status, melanoma stage, number of metastasis sites, lines of treatments, etc. Complete/partial response and stable disease were defined as class 1, while progressive disease was defined as class 2. The algorithm performances were evaluated on the test set by the percentage of treatments correctly classified in class 1 or 2. The authors of this scientific paper identified and validated, for both types of treatment (immunotherapy and targeted therapy), four predictive algorithms for drug response. These machine learning models confirmed, for melanoma, the validity of several predictor variables of response to treatments, already found in the literature. Shofty et al. [69] designed and analyzed machine learning methods for genetic background analysis in melanoma brain metastases. Indeed, genetic characterization of brain metastases, to date, is only possible by tissue analysis, which may lead to risks and complications. This study made it possible to identify the mutational status of brain metastases (specifically BRAF status) by using software that exploited conventional MRI images and correlated them with BRAF mutations. Specifically, the preoperative MRI analysis of 53 patients with melanoma brain metastases (with or without BRAF mutations) was used to predict the mutational status of the gene by analyzing 195 radiomic features associated with BRAF status. The data obtained had a mean accuracy = 0.79 ± 0.13, mean precision = 0.77 ± 0.14, mean sensitivity = 0.72 ± 0.20, and mean specificity = 0.83 ± 0.11, confirming that radiomics-based genetic characterizations can be viable alternatives to in situ biopsies. One of the questions that is still difficult to answer in melanoma, but not only melanoma, concerns predicting the development of metastasis in a specific patient. Mancuso et al. [70], in a recent study, exploited the enormous potential of machine learning to identify possible markers that can be associated with disease progression as the early stages of melanoma. The authors initially recruited 448 melanoma patients, 323 of whom were diagnosed with stage I-II disease. By means of ELISA assays, they assessed the expression levels of known cytokines involved in metastasis and, using machine learning and Kaplan-Meier techniques, analyzed the data obtained to define an algorithm capable of stratifying patients enrolled in the study according to their risk (low or high) of developing metastasis. The data obtained were analyzed using machine learning and Kaplan-Meier techniques to define an algorithm capable of accurately classifying early-stage melanoma patient with high and low risk of developing metastasis. The results obtained correlated serum levels of dermicidin (DCD), interleukin-4 (IL-4), and granulocyte-macrophage colony-stimulating factor (GM-CSF) with an increased probability of developing metastasis. Predicting the outcome of clinical treatment would greatly help the work of clinicians who could better stratify their patients to whom they would target specific and diversified therapies. One way to proceed, according to Johannett et collaborators [71], involves the analysis of patients’ histologic characteristics as potential markers for predicting response to a given treatment. Specifically, the authors developed a platform in which clinical data were integrated by deep learning on histologic samples to predict, in metastatic melanoma, response to immunotherapy. Obviously, to obtain such results, the study had to involve a training cohort of patients and a validation cohort. The classifier stratified patients into high and low risk of disease progression and, indeed, patients with a higher risk of PFS than those classified as lower risk were examined [71]. ML was also exploited in another study that identified a composite cytokine signature at baseline, associated with OS in metastatic melanoma. In this study, Wang et al. [72] used machine learning models to analyze clearance data and cytokine levels of patients with metastatic melanoma treated with nivolumab in two phase III trials. Prediction of clearance (high vs. low) by cytokine signature was significantly associated with OS in all two studies (*p* < 0.01), regardless of treatment (nivolumab vs. chemotherapy). For advanced melanoma, but in general for solid tumors, the outcome of treatment is mainly evaluated by observation of the reduction or non-reduction of the tumor mass. A very recent study suggests, however, that radiomics imaging could provide the clinician with additional and more detailed information (of mass size alone) that could help in patient management [73]. Indeed, according to the study authors, being able to accurately predict therapeutic survival benefit could also improve the therapy selection. In this study, CT images at baseline and at first follow-up of patients with metastatic melanoma, and who were treated with immunotherapy, were analyzed by exploiting radiomics and machine learning techniques. The metastatic sites detected were lymph nodes (53.0%), lungs (38.5%), liver (19.0%), and adrenal gland (11.2%). The radiomic signature arose from the set of four imaging features and, in the validation set, exceeded in predictions the standard method based on tumor diameter (Response Evaluation Criteria in Solid Tumors 1.1 (RECIST 1.1)). The results obtained, therefore, demonstrated that the in-depth analysis of a radiomic signature could predict response to treatment (in this case to pembrolizumab) in advance, giving physicians the ability to evaluate the use of alternative therapies at an early stage [73].

## 5. Future Perspectives from Artificial Intelligence in Melanoma

To date, the information we have about the characteristics of a tumor is numerous, partially due to the development of increasingly sensitive and precise technologies. Omics sciences provide a comprehensive view of the cancerous mass from molecular, metabolic, and imaging perspectives, leading to a deeper knowledge of the pathology. Melanoma is one of the cancers that, given its characteristics, has benefited from the development of these new technologies. The next step will be the reprocessing of all the data obtained through the use of complex AI software and is related to the ability that machines have to “learn” from experience. Information sharing and the ability of machines to quickly interpret data will allow the creation of precise treatment protocols. Treatment times will be able to be modulated by the clinician based on patient characteristics, experience, and based on information obtained and processed by advanced software.

## 6. Conclusions

Erroneous choices in terms of treatment approaches may affect the patient’s quality of life, which is incurred in side effects (often avoidable if unnecessary). In addition, the great expectations placed on treatments not functioning, from a psychological point of view, could negatively impact the normal conduct of a patient’s life. AI’s main purpose is to improve human performance. Indeed, the machine, after learning information from massive data pools, is able to apply the acquired knowledge to new data never processed before [74]. This highly intuitive and revolutionary concept enables, and will enable, a more precise and rational approach in many contexts of daily life, including research and health. Obviously, this will result in improvements in administration, timing, and costs related to patient management. While this revolutionary approach to study in medical practice promises to simplify the diagnostic and treatment process, it also needs to be further validated through the development of specific and pathology-dependent studies. In addition, the incorporation of all these complex analyses into clinical practice could certainly implement the myriad of information we have for individual diseases, contributing to the creation of up-to-date databases aimed at stratifying patients and assisting the clinician in the most appropriate diagnosis and treatment choice for the individual patient. Despite the immense potential of AI, to date its applications in oncology are still in the early stages and have not yet undergone major validation processes [75]. Although, as discussed in this review, there are many possible applications for melanoma, breast, lung, and prostate cancers, which represent the tumor forms now receiving the greatest benefits from AI-based devices [76]. With important early goals achieved, new further research is needed to ensure the analytical and clinical validity of AI approaches [77].

## Figures and Tables

**Figure 1 cells-11-03965-f001:**
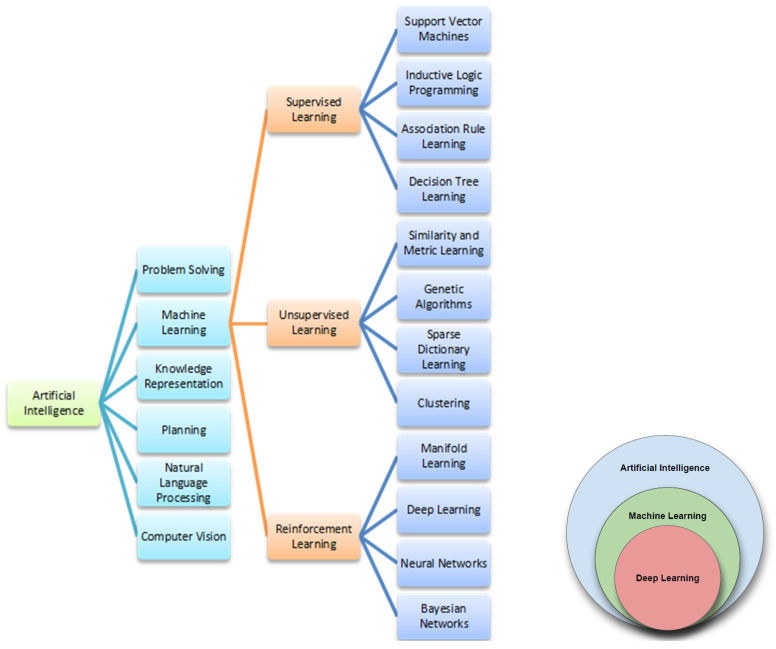
AI main fields are depicted in Figure 1. Machine learning is the heart of AI, and its more promising research area is represented by so-called deep learning, an evolution of the artificial neural network basic approach.

**Figure 2 cells-11-03965-f002:**
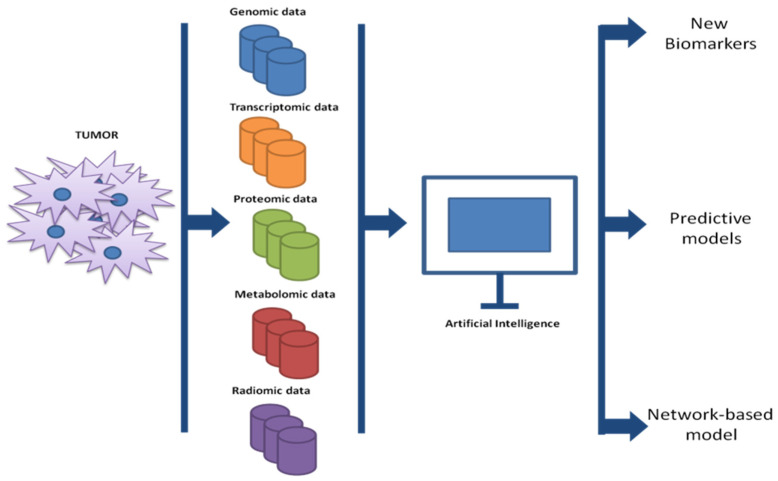
Schematic representation of biobanking and analysis of cancer data.

**Table 1 cells-11-03965-t001:** Schematic representation of the principal melanoma immunotherapeutic agents.

Inhibitor	Target	Class	Reference(s)
Ipilimumab (MDX-010)	CLT-4	Selective human IgG1 monoclonal antibody	[20]
Nivolumab (BMS-936558, MDX-1106)	PD-1	Selective human IgG4 monoclonal antibody	[21]
Pembrolizumab (MK-3475)	PD-1	Selective humanized IgG4 monoclonal antibody	[22]
Pidilizumab (CT-011)	PD-1	Selective humanized IgG1 monoclonal antibody	[23]
BMS-936559 (MDX-1105)	PDL-1	Selective human IgG4 monoclonal antibody	[24]
Atezolizumab (MPDL3280A)	PDL-1	Selective humanized IgG1 monoclonal antibody	[25]
Durvalumab (MEDI4736)	PDL-1	Selective humanized IgG1 monoclonal antibody	[26]
Avelumab (MSB0010718C)	PDL-1	Selective humanized IgG1 monoclonal antibody	[27]

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
