# Peer review of "Artificial Intelligence and Advanced Melanoma: Treatment Management Implications"

_cells, 2022, doi:10.3390/cells11243965_

Round 1
Reviewer 1 Report
The authors have provided a good overview of the current field of the use of AI in melanoma treatment. Regarding the manuscript, I have one comment as below
The authors created Figure 2 to illustrate how a multiple different datasets can be curated into an AI model as highlighted in the manuscript. I think it would be better to elaborate on the next stage further: how the developed AI model can help for predictive modeling/ network-based model. One way to study oncology/personalized cancer treatment is through mathematical modeling ( for example, The prisoner’s dilemma as a cancer model - PMC (nih.gov)). It would be very interesting for readers to read abouut how the AI model can complement or improve a mechanistic model .
Author Response
We thank the reviewer for positive comments on the manuscript. The reviewer highlighted the need to describe in the text the possible practical implications (through the use of mathematical models) that AI could bring to oncology, in terms of predicting disease progression and response to treatments. Hoping that we have captured the sense of what the reviewer asked us to do, we have included in the text what was requested.
Reviewer 2 Report
The article is very interesting before, it could be considered for further publication I have some major quires which authors need to incorporate in the revised version of Manuscript.
Abstract section does not give proper information. The introductory part needs to be rephrased. Also, highlight essentialities and future perspectives of the study.
In the introductory section,
The second para seems incorrect. Authors have written “we will focus particular attention on the novel” To me, it seems incorrect. Similarly, “Indeed, we will analyze” needs correction.
New frontiers in melanoma treatment heading
Authors have written “Thus, the need to identify new biomarkers (molecular or imaging) that are potential targets for new therapies or prognostic factors of response to treatments”…the sentence seems incomplete…
Table 1:
It is better to add references to literature of any suitable sources from where the information was retrieved.
Artificial intelligence in oncology heading
Authors have mentioned “In Figure 1 the AI organization is represented.” It is better to directly refer Fig. 1 in text at suitable place. In the same para, authors wrote “Shofty et al. designed and analyzed followed by other study ….Mancuso et al., in a recent study” and no references added. The same mistake has been repeated again and again throughout the manuscript. Also, a single para spreads over 2 pages….Please divide it into different paras.
Metastatic melanoma management heading
Authors have started a para “Goussalt et al. have developed and validated ……..” To my surprise no reference has been added to this statement. Additionally, the para is large, it is better to divide it…
Future perspectives
Intended meaning not clear, please rephrase.
Conclusion
Please update your conclusion in the light of recent reports from 2019-22.
Author Response
We thank the reviewer for considering the manuscript interesting and valid for publication. We have made all requested changes in the text:
# Abstract section was ri-modulated;
# in the introduction, we have corrected the incorrect word "novel" to "new" and replaced the word "analyze" with "describe";
#in the section “New frontiers in melanoma treatments: an overview”, we have modulate the final sentence in a more appropriate form ;
#in Table 1 the relative references was inserted;
#the bibliografy in the text was modified as request and the Figure 1 inserted in the text, when citated; the paragraph was divided;
#We agree with the reviewer that the future perspectives should be modified. The paragraph was modulated;
#as requested, conclusions were supplemented with more recent information.
Reviewer 3 Report
The authors present a brief review of the application of AI to melanoma management. It is pleasant to read and covers the main aspects. It an be accepted for publication.
Author Response
We thank the reviewer for positive comments on the manuscript. and we are glad to know that the manuscript was of interest to you for publication. The reviewer may still view the manuscript with the changes requested by the other reviewers.